

# The resistance of quantum entanglement to temperature in the Kugel–Khomskii model

Valerii E. Valiulin[1,2]*, Andrey V. Mikheyenkov[1,2],
Nikolay M. Chtchelkatchev[1,2] and Kliment I. Kugel[3,4]

**1** Moscow Institute of Physics and Technology (National Research University),
Dolgoprudny 141701, Russia
**2** Vereshchagin Institute of High Pressure Physics, RAS, Moscow (Troitsk) 108840, Russia
**3** Institute for Theoretical and Applied Electrodynamics, Russian Academy of Sciences,
Moscow 125412, Russia
**4** National Research University Higher School of Economics, Moscow 101000, Russia

* valiulin@phystech.edu

## Abstract

The Kugel–Khomskii model with entangled spin and orbital degrees of freedom is a good testing ground for many important features in quantum information processing, such as robust gaps in the entanglement spectra. Here, we demonstrate that the entanglement can be also robust under effect of temperature within a wide range of parameters. It is shown, in particular, that the temperature dependence of entanglement often exhibits a nonmonotonic behavior. Namely, there turn out to be ranges of the model parameters, where entanglement is absent at zero temperature, but then, with an increase in temperature, it appears, passes through a maximum, and again vanishes.

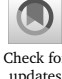
# 1 Introduction

The rapid development of quantum informatics and research on cold atoms at optical lattices gave an impetus to reboot the theory of spin–orbital ordering in solids on new physical grounds. Quantum algorithms [1–7] and quantum computing [8–12] need entanglement. Other impressive aspects of the modern entanglement problem have been intensively studied recently [13–22].

Quantum entanglement can be achieved both for identical degrees of freedom, and for fundamentally nonidentical ones. The latter situation conveniently arises in spin–orbital physics characteristic of transition metal compounds and, as recently shown, for ultracold atoms in optical traps, where the spin and orbital (or another pseudospin variable) quantum degrees of freedom coexist and can interact with each other in a controlled manner described by the Kugel–Khomskii model [23–31], which leads to their entanglement [32, 33]. However, this does not occur for all values of the interaction parameters. A significant effect of applied fields in controlling the entanglement has also been demonstrated [34]. It should be mentioned here that since the nature of the model variables can be quite different, the applied fields can also differ: from magnetic (in transition metal compounds) and electric (in optical traps) to elastic stress fields.

The ground state has been analyzed for different variants of the Kugel–Khomskii model — symmetrical SU(2) × SU(2) [34–36], SU(2) × XY [37], SU(2) × XXZ [38], and other related models [39–43].

Of course, the model under study can in some sense be related to spin chains and to the Hubbard model. Both have an extensive bibliography dealing with the study of entanglement (including its temperature evolution, see e.g. [44–62]). Note, that only few of them deal with temperature effects. Among these works, using different measures of entanglement, there are both numerical and analytical ones, and even an experiment. However, the present spin–pseudospin model is distinguished from the Heisenberg spin chain (in this case, the ladder) by a fundamentally different type of intersubsystem interaction and the independent tuning of exchange parameters in the subsystems. As for the relationship with the Hubbard model. The symmetric spin-pseudospin model under study here, under certain restrictions on the coefficients, can be obtained from the two-band Hubbard model and is very far from its standard one-band version.

The effect of temperature on entanglement in these models has been studied much less. The temperature effects on the entanglement (including non-monotonicity) have been studied mainly for two or three qubits (spins etc.), where $T$ is the bath temperature [63–69]. In some few-particle works it is noted that the cause of the nonmonotonicity is the following. The ground state is non-entangled, though the excited ones are entangled. Here we reveal the analogous effect, but for the entanglement between two multiparticle subsystems.

Temperature destroys various types of quantum long-range ordering, and this effect is especially pronounced in low-dimensional systems. Intuitively, one might expect that entanglement — a nonlocal characteristic of quantum correlations in a system — is also rapidly destroyed by temperature. However, as shown below, even for the standard symmetric spin–orbital (or in more general terms spin–psesudospin) model, this is not always the case.

In some few-particle works is noted, that the reason of the non-monotonicity is the following. The ground state is non-entangled, though the excited ones are entangled. In fact, we demonstrate the similar effect, but for many-particle and very important model.

In the this work, we investigate the evolution with temperature of the entanglement in the spin–pseudospin model. We represent a general picture of the temperature effects on the entanglement, putting the main emphasis on two following important results.

**i.** The non-monotonic temperature dependence of the entanglement. In certain ranges

of the parameters, the entanglement is absent at zero temperature, then with an increase in temperature, it appears, exhibits a peak and, eventually, vanishes again.

**ii.** Temperature robustness of the entanglement. There exists a wide set of the model parameters, for which the entanglement is almost independent of temperature within a broad temperature range. In the typical evolution of the entanglement with temperature, we can distinguish two modes: the entanglement is nearly constant for low enough temperatures and decreases at higher $T$. Note that the length of such constant mode can be rather large.

Both features reveal a significant difference between the temperature evolution of the entanglement and that of the conventional indicators of the ordering, such as average spin or suitable correlation functions.

We consider as well the effect of the applied fields on the entanglement, which sometimes turns out to be rather nontrivial.

## 2  Methods

We consider the Kugel–Khomskii model, see below Eqs. (1) and (2), with the conventional symmetric spin–pseudospin interaction (3) for a small linear cluster. We accurately determine the many-particle finite-temperature density matrix by the exact diagonalization of Hamiltonian (1). The maximum cluster size is limited by computing resources, mainly by the RAM size. We study both the cases of zero field and strong applied field in each subsystem, see below Eq. (4), in particular, the staggered (checkerboard type) field. This leads to a nontrivial and unexpected temperature evolution of the entanglement between spin and orbital degrees of freedom.

Hereinafter, we consider the chain with spin and pseudospin at each site with open boundary conditions. We calculate the complete basis of the Hamiltonian eigenvectors and the complete set of the system states by the exact diagonalization method [70–74]. This leads to the estimation of the finite-temperature density matrix (5). Any measure of the entanglement can be obtained based on this matrix, in particular, negativity (more exactly, logarithmic negativity) [75–79]. The Hamiltonian matrices for the systems under study are very sparse, so it is natural to use the sparse matrix format. As it was mentioned above, the maximum available size of the chain for comprehensive calculation is determined by the computational resources, in fact, both by the RAM size and the CPU hours, so we extrapolate the results to $1/N \to 0$.

One of the most suitable packages here is the QuTiP, which simplifies the work with quantum objects, and requires relatively low computation time in comparison to the others quantum computation solutions [80,81]. The package has a very handful interface for constructing the many-particle Hamiltonian, its complete basis, and subsequent manipulation, using a vast number of quantum operators. Every object in the package is by default converted to sparse format, which significantly simplifies further processing. Thus, all our quantum computations were performed in the QuTiP package.

## 3  Model system

### 3.1  Microscopic Hamiltonian

The microscopic description of hybrid spin and pseudospin models usually starts from the Hubbard model, which is relevant to strongly interacting quantum particles on a lattice. These particles can be represented either by electrons in transition metal compounds, for which pseudospin is provided by different atomic orbitals, or by ultracold atoms with the Bose or

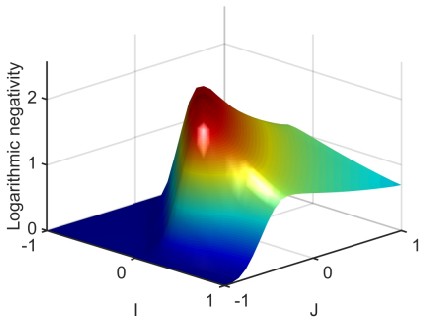 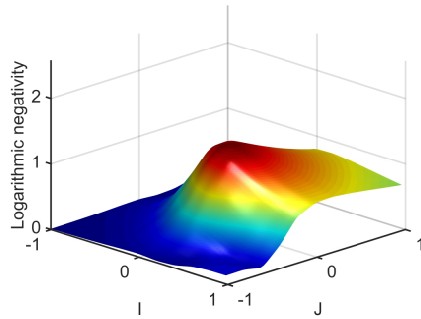

Figure 1: General view of the temperature evolution the entanglement (logarithmic negativity) for $K = -1$ at zero applied field. Left panel — low-temperature regime, $T = 0.005$; right panel — high-temperature regime, $T = 0.15$.

Fermi statistics on optical lattices, where pseudospin, e.g., can be related to the type of an atom occupying a lattice site. In any case, when the on-site Hubbard interactions dominate over the hopping between lattice sites, the Hubbard model can be reduced to the spin–pseudospin Kugel–Khomskii Hamiltonian.

We focus here on the typical version of the Kugel–Khomskii model — the SU(2)×SU(2) model with SU(2) symmetries for both spin-1/2 and pseudospin-1/2 operators ($\hat{\mathbf{S}}$ and $\hat{\mathbf{T}}$). We also introduce different kinds of applied fields and study their effect on the entanglement.

The Hamiltonian of the model reads

$$\widehat{\mathbf{H}} = \widehat{\mathbf{H}}_s + \widehat{\mathbf{H}}_t + \widehat{\mathbf{H}}_{ts}. \tag{1}$$

Here $\widehat{\mathbf{H}}_s$, $\widehat{\mathbf{H}}_t$ are Heisenberg-type interactions in the spin pseudospin–spin subsystems:

$$\widehat{\mathbf{H}}_s = J \sum_{<i,j>} \widehat{\mathbf{S}}_{\mathbf{i}} \widehat{\mathbf{S}}_{\mathbf{j}}, \qquad \widehat{\mathbf{H}}_t = I \sum_{<i,j>} \widehat{\mathbf{T}}_{\mathbf{i}} \widehat{\mathbf{T}}_{\mathbf{j}}, \tag{2}$$

and $\widehat{\mathbf{H}}_{ts}$ is the interaction between subsystems. For the symmetrical model, it has the form

$$\widehat{\mathbf{H}}_{ts}^{(1)} = K \sum_{<i,j>} \left( \widehat{\mathbf{S}}_{\mathbf{i}} \widehat{\mathbf{S}}_{\mathbf{j}} \right) \left( \widehat{\mathbf{T}}_{\mathbf{i}} \widehat{\mathbf{T}}_{\mathbf{j}} \right). \tag{3}$$

In Eqs. (2) and (3), $\mathbf{i}$ and $\mathbf{j}$ are vectors denoting the positions of the nearest neighbors, $\widehat{\mathbf{S}}_{\mathbf{i}}$ and $\widehat{\mathbf{T}}_{\mathbf{i}}$ are spin and pseudospin operators, the latter being related to the orbital degrees of freedom. We consider the common case when $S = 1/2$, $T = 1/2$.

The additional terms to the Hamiltonian describing the effect of external fields in both subsystems can be written as

$$\widehat{\mathbf{H}}_f = -\mathcal{H}_s \sum_{\mathbf{i}} \widehat{\mathbf{S}}_{\mathbf{i}}^z - \mathcal{H}_t \sum_{\mathbf{i}} \widehat{\mathbf{T}}_{\mathbf{i}}^z, \tag{4}$$

where $\mathcal{H}_s$ and $\mathcal{H}_t$ are fields in spin and pseudospin systems, respectively. As it was mentioned, the actual magnetic fields in the model are in fact, not necessarily magnetic ones, but can have different physical nature. In particular, that is why, the staggered fields are possible.

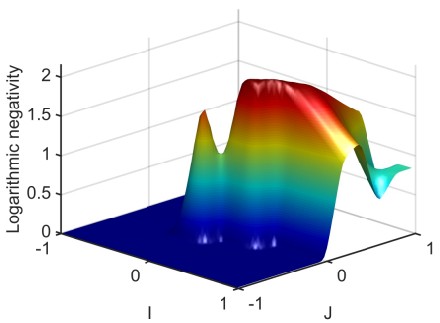 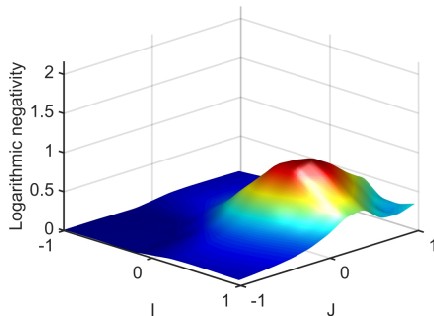

Figure 2: General view of the temperature evolution the entanglement (logarithmic negativity) for $K = +1$ at zero applied field. Left panel — low-temperature regime, $T = 0.005$; right panel — high-temperature regime, $T = 0.15$.

## 3.2 Temperature dependent quantum correlations

A sufficiently high applied magnetic field (4) obviously suppresses the entanglement. However, in the most physically interesting range of parameters (all the exchange integrals $J, I, K$ and temperature $T$ are of the same order of magnitude) entanglement is not suppressed. Moreover, even at $T = 0$ the applied field can lead to an increase in the entanglement [34]. For nonzero temperatures, the behavior of entanglement can be also counterintuitive in some cases.

As it was mentioned, we investigate the entanglement between spin and orbital degrees of freedom at finite temperature, admitting the possibility of nonzero applied fields in both subsystems. In this case, the finite-temperature density matrix can be written as follows [77, 78]

$$\rho(T) = Z^{-1} \sum_i \rho_i \exp(-E_i/T), \tag{5}$$

where $Z$ is the standard partition function, the sum runs over all possible states of the system characterized by partial density matrices $\rho_i$ for the temperature $T$ with energy $E_i$.

We use the widespread and one of the most convenient measures of entanglement at finite $T$ — the logarithmic negativity (LN) [76]

$$LN(\rho(T)) = \ln(||\rho^{T_i}||), \tag{6}$$

where $\rho^{T_i}$ is the partial transpose of $\rho$ with respect to subsystem $i$ (being either spin or pseudospin one), $||X|| = Tr\sqrt{X^\dagger X}$ is the trace norm of the operator $X$, and $\ln(a)$ is the natural logarithm of $a$. For the problem in hand, the LN is quite practical, it is easy to evaluate, it is equal to zero in the absence of entanglement (with exception of some low-dimensional cases), and is consistent with other entanglement measures, such as concurrence [75] at zero temperature limit.

So, we first numerically estimate Hamiltonian matrix, then use the exact diagonalization method (implemented in QuTiP) to obtain the set of energy levels and wave functions. This makes it possible to construct the finite-temperature density matrix of the system. Then we estimate is the partial transpose of the calculated matrix with respect to one of the subsystems (either spin or pseudospin). The calculation of the trace norm and its logarithm completes the $LN(\rho(T))$ estimation.

A standard evaluation for a chain of seven cites for a particular temperature takes about several days, utilizing parallel computations. For the sake of the reliability, we have computed several point of interest (it took about a week) for longer chains with $N = 8, 9,$ and $10$. The

results differ qualitatively only slightly, and, as it was mentioned above, allow for a good extrapolation to $1/N \to 0$. When it was possible, we compared our results with those from other publications on the entanglement.

# 4 Entanglement temperature evolution

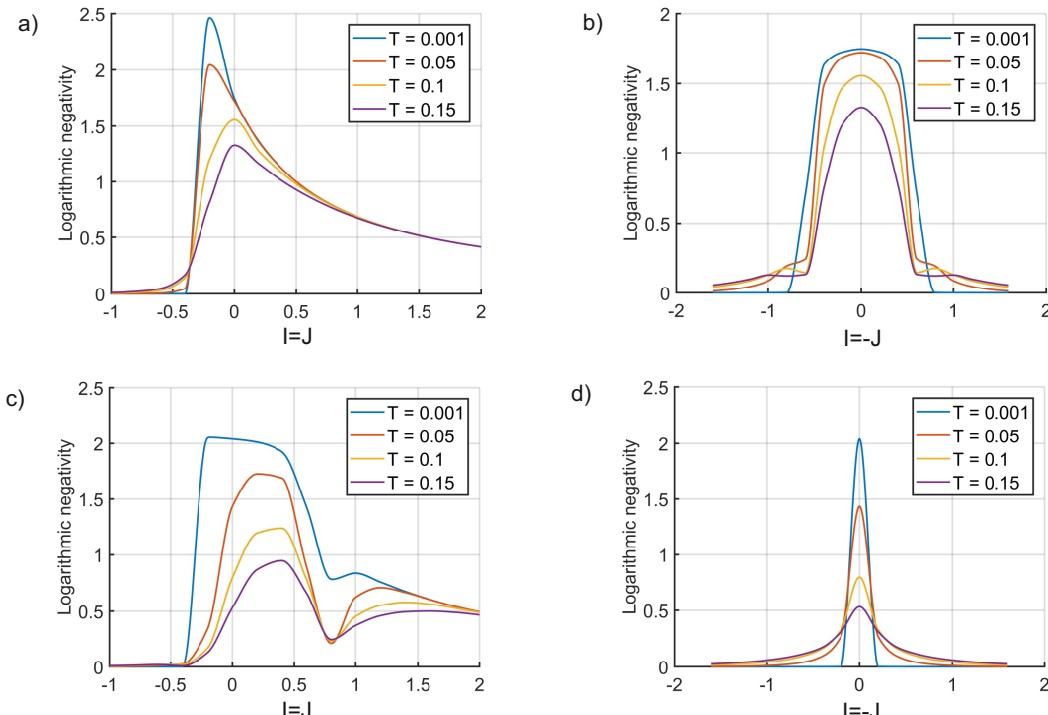

Figure 3: **(a)** Temperature evolution of the entanglement for the main diagonal of the $I - J$ plane ($I = -1, J = -1$)-($I = +1, J = +1$), and negative intersubsystem exchange $K = -1$. Two vivid effects are clearly seen. The first one — different rates of the entanglement decay with temperature near the maximum and far from it (and even the resistance of entanglement to temperature for $I = J \gtrsim 1$). The second — the non-monotonic behavior of the entanglement with temperature to the left of the maximum. **(b)** The same for the antidiagonal of the $I - J$ plane ($I = -1, J = +1$)-($I = +1, J = -1$). Difference of the entanglement decay rates is visible, though less, than in panel (a). The non-monotonicity of the temperature dependence is obvious on both sides of low-temperature entanglement 'cliff'. **(c)** Temperature evolution of the entanglement for the main diagonal of the $I - J$ plane ($I = -1, J = -1$)-($I = +1, J = +1$), and positive intersubsystem exchange $K = +1$. The non-monotonicity of the entanglement with temperature is hardly distinguishable, but can be detected in two different regions: to the left of the maximum and for $I = J \sim 0.8$. The common asymptotics of the curves is seen for large $I, J$. **(d)** The same for the antidiagonal of the $I - J$ plane ($I = -1, J = +1$)-($I = +1, J = -1$). One can see considerable temperature growth of the entanglement at both sides of the maximum, where at low temperature it is absent.

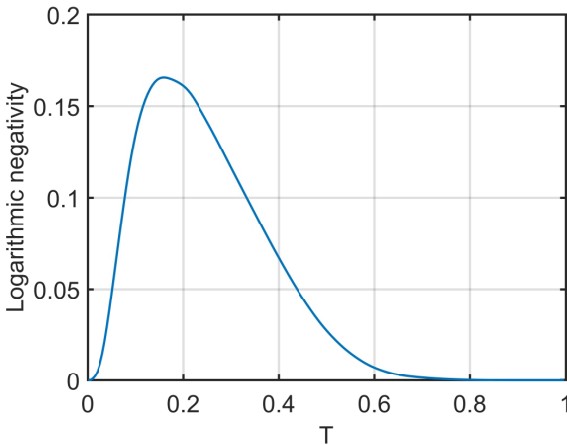

Figure 4: Non-monotonic behavior of the entanglement with temperature at the point $I = J = -0.4$ near the low-temperature sharp peak ($K = -1$). See also Fig. 3(a).

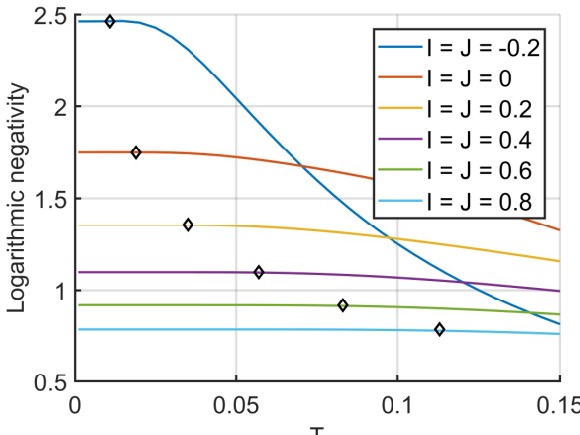

Figure 5: Temperature dependence of the entanglement at some points of the main $I - J$ plane diagonal with $K = -1$. For any point it exhibits two modes: constant one for low enough temperatures and decreasing for high $T$. The mode crossover points are marked by diamonds. The higher are the $I, J$ values, the larger is the duration of the constant mode. See also Fig. 3(a).

From the first glance, the temperature dependence of the entanglement seems to be obvious — the temperature necessarily washes the entanglement out. Indeed, Figs. 1 and 2, which illustrate the negativity-measured entanglement at low and high temperatures (at the inter-subsystem exchange parameter $K = \pm 1$) lend support to the above statement. At any point of $I - J$ plane, the entanglement either remains zero, or if it was nonzero for $T = 0$, it tends to zero at sufficiently high temperatures.

However, a more detailed analysis radically changes the matter.

## 4.1 Zero field, $K = -1$

In Fig. 3(a), we represent the evolution of entanglement at moderate temperatures for the main diagonal in the $I - J$ plane ($I = J$), and negative intersubsystem coupling, $K = -1$. Two unexpected effects are clearly seen.

The first one — different rates of the entanglement decay with temperature near the maximum and far from it $|\frac{dLN(T)}{dT}(I = J \approx -0.2)| \gg |\frac{dLN(T)}{dT}(I = J \approx +0.5)|$. One can find that the entanglement even resists to temperature for $I = J \gtrsim 1$, when the entanglement remains constant (and nonzero) in a wide temperature range.

The second effect is the temperature non-monotonicity in the temperature evolution of the entanglement to the left of the maximum. At $T = 0$, the entanglement to the left of the 'cliff' is zero, but at $T > 0$, it appears and grows with temperature (it will subsequently pass through a maximum, and again decrease to zero, see more details in Fig. 4 (this effect was predicted based on the other approach in Ref. [82]).

Both these effects are also present at the antidiagonal of the $I - J$ plane ($I = J$) with $K = -1$, see Fig. 3(b). There we have different rates of the temperature-induced decay of the entanglement at different points in the $I - J$ plane. An interesting limiting case here corresponds to the vicinity (both sides) of the maximum, where the entanglement exhibits a resistance to temperature and a non-monotonic dependence on temperature.

Reproducing the complete 3D temperature dependence of the entanglement requires fantastically large computational resources. Taking the symmetry into account reduces them only by a factor of two. However, two cuts of a relatively smooth 3D plot by diagonal and antidiagonal planes obviously well reproduce the general picture. So these two cuts are sufficient to identify all the basic patterns. Hereafter, for other particular cases, we follow the foregoing scheme (note, that Fig. 1 and Fig. 2 are reconstructed from diagonal and antidiagonal data points).

## 4.2 Zero field, $K = +1$

Now, we turn to the version with the positive intersubsystem exchange $K = +1$. The low-temperature entanglement landscape substantially differs from that for $K = -1$, compare Figs. 1(a) and 1(b) (see also the discussion concerning the case for $T = 0$ in Ref. [34]). Nevertheless, all the effects, considered in the previous subsection, also manifest themselves here.

In Fig. 3(c), we represent the temperature evolution of the entanglement for the main diagonal of the $I - J$ plane ($I = J$) and $K = +1$. Figure 3(d) corresponds to the antidiagonal of the $I - J$ plane ($I = -J$).

On the main diagonal the non-monotonicity in the temperature dependence of the entanglement is hardly distinguishable, nevertheless it can be revealed in two different ranges of parameters: to the left of the maximum and for $I = J \sim 0.8$. The resistance of entanglement to temperature manifests itself as the common asymptotics of the curves at large $I$ and $J$.

Such resistance is undistinguishable at the antidiagonal, but two other features are clearly visible there — different rates of the entanglement decay with temperature at different points in the $I - J$ plane and considerable temperature growth of the entanglement on both sides of the maximum, where at low temperatures, it is absent.

As a result, we arrive at the following conclusion. The initial state at $T = 0$ for the system with the positive intersubsystem exchange $K = +1$ is quite different from that for $K = -1$. Nevertheless, with the growing temperature, we again obtain the same surprises related to the entanglement — the non-monotonicity and resistance to temperature.

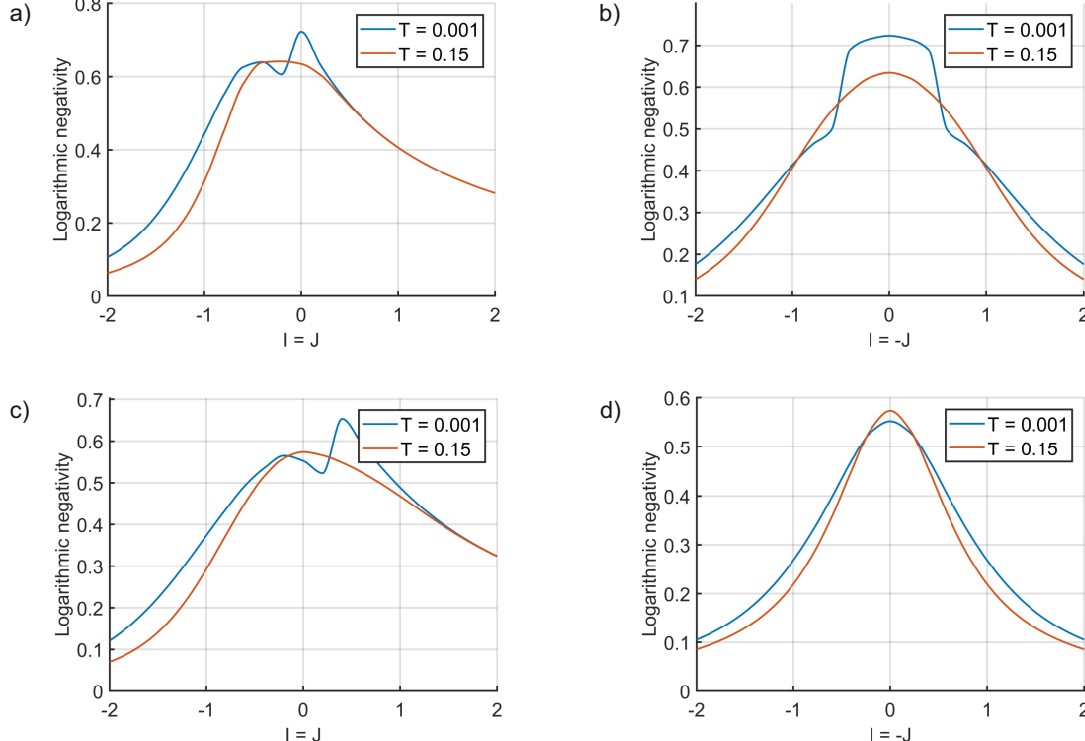

Figure 6: **(a)** The evolution of the entanglement with temperature for the main diagonal of the $I-J$ plane, and negative $K = -1$ in the presence of staggered fields in both subsystems. The low-$T$ two-peak structure caused by the concerted action of the intersubsystem exchange and the staggered fields erodes with the increasing $T$. At high $T$ one broad maximum is formed. Compare Fig. 3(a) without fields. **(b)** The same for the antidiagonal of the $I-J$ plane. The non-monotonic behavior of the entanglement with temperature is seen at both sides of the main peak. Compare Fig. 3(b) at zero field. **(c)** Evolution of the entanglement with temperature for the main diagonal of the $I-J$ plane, and positive $K = +1$ in the presence of staggered fields in both subsystems. The shape of this plot is similar to that in Fig. 6(a): the low-$T$ two-peak structure erodes with the increasing $T$, forming one broad peak. Compare to Fig. 3(c) at zero field. **(d)** The same for the antidiagonal of the $I-J$ plane. In contrast to Fig. 6(b), the non-monotonic behavior of the entanglement with temperature is seen just at the peak. Compare to Fig. 3(d) at zero field.

## 4.3 Non-monotonicity and resistance to $T$

In this subsection, we investigate two aforementioned effects in more detail.

The first one is the non-monotonicity of the temperature dependence of the entanglement in particular areas in the $I-J$ plane. Such areas are located mainly near the low-temperature peak in the entanglement. In Fig. 4, we show the entanglement in a wide temperature range at one of these points, specifically, at the $J = I = -0.4$ point (with $K = -1$). The entanglement is definitely absent at zero temperature, then, with an increase in temperature, it arises, passes through a maximum, and again vanishes at $T \sim 1$. The initial part of this curve can be recognized in Fig. 3(a).

The same effect is reproduced by the calculations up to high enough $T$ in all areas of Fig. 3, which exhibit the non-monotonicity in the temperature dependence of the entanglement — definite or barely noticeable. We will not illustrate this fact since it is rather obvious.

The second finding is the resistance of the entanglement under effect of temperature in the wide temperature range in particular areas of the $I-J$ plane. Areas of this kind are located near the main diagonal on the descending part of the plots illustrating the temperature dependence of the entanglement and are seen in Figs. 3(a) and (c).

Fig. 5 illustrates this statement in more detail. We plot the temperature dependence of the entanglement at the chosen points in the main diagonal of the $I-J$ plane (with $K = -1$). For any point, two modes are seen: a constant at low enough temperatures and a decreasing branch at high $T$. The higher are the $I, J$ values, the larger is the duration of the constant mode. For large enough $I = J$, there exists a wide temperature range with almost constant entanglement.

The thorough calculations show that for $K = -1$, the effect is also preserved around the main diagonal and in the corresponding area of Fig. 3c ($K = +1$).

### 4.4 Nonzero field, $K = \pm 1$

Now we turn to the case of nonzero applied fields. Several versions are possible for different realizations of the spin–pseudospin model — uniform field in one subsystem, uniform fields in both subsystems (parallel or antiparallel), staggered fields in one subsystem or in both. See the discussion of different possibilities for $T = 0$ in Ref. [34].

Here, we focus on the most interesting and nontrivial case of staggered fields in both subsystems. It is illustrated in Fig. 6(a-b) for $K = -1$ and in Fig. 6(c-d), $K = +1$. Note that in contrast to Fig. 3 with four reference temperatures $T = 0.001, 0.05, 0.1$, and $0.15$, here only two of them are present — $T = 0.001$ and $T = 0.15$ , because the plots are quite similar.

In Fig. 6(a), we represent the evolution of entanglement with temperature ($K = -1$) in the presence of staggered fields in both subsystems for the main diagonal of the $I-J$ plane. Here, the low-$T$ two-peak structure caused by the concerted effect of the intersubsystem exchange and the staggered fields erodes with an increase in $T$. At high $T$, one broad maximum is formed. The resistance of entanglement to temperature for $I = J \gtrsim 0.5$ is clearly seen (compare to Fig. 3(a) at zero field).

In Fig. 6(b), we show the same dependence for the antidiagonal of the $I-J$ plane ($I = -J$). The non-monotonicity of the temperature dependence of the entanglement is clearly seen for both sides of the main peak (compare to Fig. 3(b) at zero field).

Figures 6(c) and 6(d) correspond to the last case under study — staggered fields in both subsystems with $K = +1$. In Fig. 6(c), we illustrate the evolution with temperature of the entanglement for the main diagonal of the $I-J$ plane. It looks like the corresponding picture for negative intersubsystem exchange $K = -1$ (Fig. 6(a)) except a small rescaling. The low-$T$ two-peak structure erodes with the increasing $T$ forming one broad peak. The entanglement is resistant to temperature for $I = J \gtrsim 1$ (compare to Fig. 3(c) at zero field).

In Fig. 6(d), we show the evolution with temperature of the entanglement for the antidiagonal of the $I-J$ plane. In contrast to the corresponding picture for negative intersubsystem exchange $K = -1$ (Fig. 6(b)), the non-monotonic behavior of the entanglement with temperature can be seen just at the peak (compare also to Fig. 3(d) at zero field).

## 5 Conclusions

In this paper, we put the main emphasis on the problem of temperature dependence of the quantum entanglement in spin–orbital (spin–pseudospin) models. As a quantitative measure of the entanglement, we have chosen the logarithmic negativity focusing on the case of finite chains described by the symmetric SU(2)×SU(2) model. The analysis was based on the exact

diagonalization technique allowing us to calculate the temperature-dependent density matrix, which in fact provides a possibility of finding out any measure of the entanglement.

The obtained results appear to be rather nontrivial: within a wide range of parameters $I$, $J$, and $K$ characterizing the spin–spin, pseudospin–pseudospin, and biquadratic spin–pseudospin interactions, respectively, the entanglement can either be nearly independent of temperature, or it can even arise within such ranges in the $(I,J)$ plane, where the entanglement is zero at $T = 0$. Note also that the entanglement is the most clearly pronounced at small values of $I$ and $J$ (as compared to $K$) and tends to zero at large absolute values of these parameters. However, the behavior of entanglement is very sensitive to the relative sign of parameters of $I$ and $J$. For $I$ and $J$ of the opposite signs (antidiagonal in the $(I,J)$ plane), the entanglement decays beginning from the $I = J = 0$ point with an increase in the absolute values of $I$ and $J$, whereas the rate of this decay becomes slower with the growth of temperature. Note, that obvious symmetry of the Hamiltonian (1) is clearly seen in all the figures, referring to the antidiagonal.

In contrast, for $I$ and $J$ of the same sign (diagonal in the $(I,J)$ plane), the plots of entanglement as function of the absolute values of $I$ and $J$ are highly asymmetric. They exhibit a fast decay at negative $I$ and $J$, and rather slowly decrease at positive values of these parameters. In the mentioned parameter range, we we obviously detect the entanglement resistance to temperature.

All these observations suggest that the entanglement is closely related to the tendency to the formation of specific order parameters. Indeed, since the maximum entanglement is observed at relatively large absolute values of the intersubsystem exchange $K$, we can relate the entanglement with the dominant role of the spin–pseudospin correlations [82]. At the same time, at large $I$ and $J$, the spin–spin, pseudospin–pseudospin, or both correlation functions begin to dominate, thus destroying the entanglement between spin and pseudospin degrees of freedom. With the growth of temperature such correlations become weaker giving rise to the possibility of entanglement in the parameter ranges, where it was suppressed at $T = 0$. Here we see another interesting effect — non-monotonic temperature behavior of the entanglement.

If $I$ and $J$ have opposite signs, this favors ferro- and antiferromagnetic correlations in the spin and pseudospin channels, or *vice versa*, so the effect of such correlation on the entanglement should be the same on both sides of the $I = J = 0$ point. Note that for negative $K$, spins and pseudospins in the corresponding term should be pairwise parallel to ensure the energy minimum implying rather strong correlations. For positive $K$, at least one spin variable should have the sign opposite to three others, hence one could expect weaker correlations in this case. In fact, such difference in the strength of correlations manifests itself in a smaller width of the entanglement peak at positive $K$ in comparison to that for negative $K$. For $I$ and $J$ of the same sign, negative values of these parameters favor the spin and pseudospin ferromagnetism, which should strongly suppress the entanglement. At the same time, positive $I$ and $J$ give rise to antiferromagnetic correlations, for which one should not expect such a pronounced effect on the entanglement. Thus, the resulting entanglement plot for $I$ and $J$ of the same sign appears to be quite asymmetric. Note in addition that the staggered field acting differently on spin and pseudospin variables could enhance the spin–pseudospin correlations, and hence the entanglement.

As far as the entanglement in many-body systems is concerned, it is usually important for finding out the range of existence for quantum phase transitions and revealing the areas in the phase diagram exhibiting the enhanced quantum fluctuations. In the spin-pseudospin models under study, it is especially important since it highlights the ranges, where the entangled spin-orbital excitations play a crucial role in the thermodynamics of the system. In such a case, the nonmonotonocity, especially, the emergence of it only at finite temperature can reveal important specific features of the thermal characteristics of the system.

The above qualitative reasoning just illustrates the rich physics involved in the temperature effects on the entanglement in spin–pseudospin models. Therefore, we believe that our work sheds additional light to still unexplored prospects in the field of quantum entanglement.

# Acknowledgments

V.E.V. and N.M.S. acknowledge the support of the Russian Science Foundation (project No. 18-12-00438) in the part concerning numerical calculations. K.I.K. acknowledge the support of the Russian Science Foundation (project No. 20-62-46047) in the part concerning the data analysis.

The computations were carried out on MVS-10P at Joint Supercomputer Center of the Russian Academy of Sciences (JSCC RAS). This work has been carried out using also computing resources of the Federal Collective Usage Center Complex for Simulation and Data Processing for Mega-Science Facilities at NRC "Kurchatov Institute", http://ckp.nrcki.ru/.

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
