# Peer review of "The resistance of quantum entanglement to temperature in the Kugel-Khomskii model"

_SciPost Physics, doi:SciPost Phys. Core 6, 025 (2023)_

## Round 1 · Referee Report · Anonymous (Referee 1) · 2022-3-31

Strengths

  1. Topical area.
  2. Clearly written.

Weaknesses

  1. Nonmonotonicity of nearest-neighbor entanglement wrt temperature was previously known. For small as well for infinite quantum spin systems. It is possible that it was not known for the model considered.

  2. These refs are not given, but more importantly, the paper does not bring anything more than the nonmonotonicity mentioned. Like, is it very important that entanglement is nonmonotonic wrt temperature in this particular spin-1/2 model? Is this model used for some task where entanglement is used as a resource and this nonmonotonicity is crucial or could potentially be crucial?

Report

Given the above points, I do not recommend the publication of this paper.

---

## Round 1 · Referee Report · Anonymous (Referee 2) · 2022-5-10

Report

The manuscript aims at studying resistance of entanglement in the Kugel–Khomskii model to temperature. The Kugel–Khomskii model is a spin–pseudospin system (the latter being e.g., electron's or vacancy's orbital degree of freedom) defined by a specific type of interaction. According to the authors, this type of interaction can occur in lattices of atoms for which the on-site Hubbard interactions are dominating. The authors start by finding eigenstates of the Kugel–Khomskii Hamiltonian numerically for chains of atoms up to 10. The authors then consider thermal states in which the density operator's dependence on temperature is explicit. The main part of the manuscript is dedicated to measuring entanglement in the thermal states of the Kugel–Khomskii model for various parameters of the Hamiltonian and different temperatures. As a main highlight of their work the authors choose non-monotonicity of entanglement with respect to temperature for certain model parameters.

I find the considered problem interesting and motivating. At the same time I have serious doubts that very similar problems have not been studied earlier. For instance, [V.E. Korepin, Phys. Rev. Lett. 92, 096402 (2004)] studied entanglement in the Hubbard model and its dependence on temperature. Unfortunately, the manuscript does not provide an adequate review of the literature and does not state explicitly in which way the findings of the current work are novel. Without such a comparison, in particular a proper review of the previous results on entanglement in the Hubbard model, I cannot recommend this work for publication.

Another major flaw in this manuscript, in my opinion, is the lack of details on how the results were obtained. For example, the Hamiltonian given by Eqs.(1-4) is very general and allows for arbitrary configurations of atoms in a lattice. At the same time, the authors state in the introduction that they study only chains. Most importantly, the authors do not provide description of their mathematical derivations since the whole analysis is numerical. In this case, however, a computer code should be provided. A few formulas included in the manuscript are not enough for evaluation of the results' correctness and in some cases are even confusing. In particular, there is something clearly missing in the description of the density operator in Eq.(5). If the density operators on the right-hand side of Eq.(5) are normalized, then the total density operator on the left-hand side is not normalized.

Overall, despite a potential interest of the SciPost readers to the considered problem, I cannot recommend the manuscript for publication in its current form.
  • validity: -
  • significance: good
  • originality: -
  • clarity: ok
  • formatting: acceptable
  • grammar: below threshold

---

## Round 2 · Referee Report · Anonymous (Referee 4) · 2022-11-25

Report
In the new version of the manuscript, the authors have taken into consideration the suggestions of both referees and have implemented a few changes in the text. The most noticeable modification is an increased number of the referenced works, which is accompanied by an increased discussion on how the model considered in the paper is different from others in the literature in the context of study of temperature effect on entanglement. Additionally, the authors have included some more details about their calculations in the new version of their manuscript.
Taking into account the implemented changes and the reply, unfortunately, I still cannot recommend the current work for publication. My main concern regarding originality of the results is still present. Perhaps, I should have stressed more in my first report on the importance of the comparison of the results of the manuscript with the results of the cited works. This comparison should not only concern the type of model considered, but also the effects that are observed there, i.e., the results themselves. For example, is it the case that nonmonotonicity of entanglement occurs exclusively in the Kugel–Khomskii model, or, perhaps, this effect is much more profound than in other systems? In a way, in their reply the authors have even confirmed that very similar problems have been studied before, and that similar effects have already been reported. For example, the authors write: "The temperature effects on the entanglement (including nonmonotonicity) have been studied mainly for two or three qubits." The authors' reply to this criticism is: "As far as we know, the analysis of the temperature entanglement and its nonmonotonicity in the spin-pseudospin model is still absent". However, it is not made clear from the text or the reply how this choice of the model affects the results, i.e., the dependence of entanglement on temperature. Is the considered model the most sensitive one or, contrary, the most robust to temperature fluctuations? Without such questions being clearly answered in the manuscript, it is impossible to judge about the impact of the current work, and hence to recommend the manuscript for publication.
Report
I have read the authors' response to the comments of both the referees. As far as I know, the dependence of the nonmonotonicity of entanglement wrt temperature was not checked before. It would still be good to know why such dependence is of interest, and this is not answered clearly in the paper. However, I think that the study could lead to such answers and hence I recommend its publication in SciPost.

---

## Round 2 · Author Response

Hereafter is the list of our corrections
The referee 1 wrote: 1. Nonmonotonicity of nearest-neighbor entanglement wrt temperature was previously known. For small as well for infinite quantum spin systems. It is possible that it was not known for the model considered. Our reply The temperature effects on the entanglement (including nonmonotonicity) have been studied mainly for two or three qubits (spins etc.), where T is the bath temperature [1-7]. In some few-particle works, it is noted that the cause of the nonmonotonicity is the following. The ground state is non-entangled, though the excited ones are entangled. There are numerous works on infinite Hubbard and spin systems, mainly chains, see e.g. [8-26]). Note that only few of them deal with temperature effects. As far as we know, the analysis of the temperature entanglement and its nonmonotonicity in the spin-pseudospin model is still absent. We are grateful to the reviewer for the useful remark, and in the revised manuscript (Version 2), we added references to some important works in this area.
The referee wrote: 2. These refs are not given, but more importantly, the paper does not bring anything more than the nonmonotonicity mentioned. Like, is it very important that entanglement is nonmonotonic wrt temperature in this particular spin-1/2 model? Is this model used for some task where entanglement is used as a resource and this nonmonotonicity is crucial or could potentially be crucial? Our reply The spin-pseudospin model considered in this paper known since the 1970s, has get a second wind after the experimental discovery of orbital waves ([27, 28]) and is now widely used. Thus, reviews of only recent works in this and closely related areas ([29, 30]) contain several hundred of references. Moreover, the model is being studied in application not only to transition metal compounds, but also to other systems with two degrees of freedom, which can be described by spin (pseudospin) operators. Such a situation is realized, for example, in some systems of ultracold atoms in optical lattices. The version of the model, for which both the spin and the pseudospin are equal to 1/2, is studied much more often than others, both due of its relative simplicity and because the field of its experimental implementation is wider (in the future, we intend to investigate entanglement in other versions of the model). Attention to the entanglement in this model is related primarily to its physical nature it is clear that the analysis of many-particle quantum entanglement between non-identical degrees of freedom (spin and orbital in the canonical implementation) is of considerable interest. A relatively outdated review of the subject ([31]) the later ones are unknown to us also includes more than 150 references. We believe that these circumstances explain the impossibility of an exhaustive bibliography within the framework of a regular article. We have tried to provide references to the most important works for the issue under study, to the most recent ones and to the key reviews. We also have not met any works on the temperature behavior of the entanglement in the spin-pseudospin model, however, we have added several references to the works where temperature effects are studied in the purely spin models. As for nonmonotonicity, we are not aware of works in the model under study where it would have been discovered (there are only indirect indications of the effect in the work of the same authors). Moreover, we point out a range of parameters where the entanglement, which is absent at zero temperature, arises and passes through a maximum at T >0. It seems that such a phenomenon should not be overlooked. We have added several extensions to the revised manuscript (Version 2) explaining the importance of the model under study with the corresponding references. As far as the entanglement in many-body systems is concerned, it is usually important for finding out the range of existence for quantum phase transitions and revealing the areas in the phase diagram exhibiting the enhanced quantum fluctuations. In the spin-pseudospin models under study, it is especially important since it highlights the ranges, where the entangled spin-orbital excitations play a crucial role in the thermodynamics of the system. In such a case, the nonmonotonocity, especially, the emergence of it only at infinite temperature can reveal important specific features of the thermal characteristics of the system. We have added a more detailed description of the subject to the revised manuscript (Version 2) in the Conclusions section.
Reply to the referee 2 The referee wrote: I find the considered problem interesting and motivating. At the same time I have serious doubts that very similar problems have not been studied earlier. For instance, [V.E. Korepin, Phys. Rev. Lett. 92, 096402 (2004)] studied entanglement in the Hubbard model and its dependence on temperature. Unfortunately, the manuscript does not provide an adequate review of the literature and does not state explicitly in which way the findings of the current work are novel. Without such a comparison, in particular a proper review of the previous results on entanglement in the Hubbard model, I cannot recommend this work for publication. Our reply The temperature effects on the entanglement (including nonmonotonicity) have been studied mainly for two or three qubits (spins etc.) where T is the bath temperature [1-7]. In some few-particle works it is noted that the cause of the nonmonotonicity is the following. The ground state is non-entangled, though the excited ones are entangled. Of course, the model under study can in some sense be related to spin chains and to the Hubbard model. Both have an extensive bibliography dealing with the study of entanglement (including its temperature evolution, see e.g. [8-26]). Note, that only few of them deal with temperature effects. Among these works, using different measures of entanglement, there are both numerical and analytical ones, and even an experiment. However, the present spin-pseudospin model is distinguished from the Heisenberg spin chain (in this case, the ladder) by a fundamentally different type of intersubsystem interaction and the independent tuning of exchange parameters in the subsystems. As for the relationship to the Hubbard model the symmetric spin-pseudospin model under study here, with certain restrictions on the coefficients, can be obtained from the two-band Hubbard model and is very far from its standard one-band version. That is why, we initially had not found it possible to mention entanglement in the Hubbard and purely spin models in the bibliography. We are grateful to the reviewer for the useful remark, and in the revised manuscript (Version 2), we added references to some important works in this area. Then, as far as we know, the analysis of the temperature entanglement and its nonmonotonicity in the spinpseudospin model is still absent. The spin-pseudospin model considered in this paper known since the 1970s, has get a second wind after the experimental discovery of orbital waves ([27, 28]) and is now widely used. Thus, reviews of only recent works in this and closely related areas ([29, 30]) contain several hundred of references. Moreover, the model is being studied in application not only to transition metal compounds, but also to other systems with two degrees of freedom, which can be described by spin (pseudospin) operators. Such a situation is realized, for example, in some systems of ultracold atoms in optical lattices. The version of the model, for which both the spin and the pseudospin are equal to 1/2, is studied much more often than others, both due of its relative simplicity and because the field of its experimental implementation is wider (in the future, we intend to investigate entanglement in other versions of the model).
The referee wrote: Another major aw in this manuscript, in my opinion, is the lack of details on how the results were obtained. For example, the Hamiltonian given by Eqs.(1-4) is very general and allows for arbitrary configurations of atoms in a lattice. At the same time, the authors state in the introduction that they study only chains. Most importantly, the authors do not provide description of their mathematical derivations since the whole analysis is numerical. In this case, however, a computer code should be provided. A few formulas included in the manuscript are not enough for evaluation of the results’ correctness and in some cases are even confusing. In particular, there is something clearly missing in the description of the density operator in Eq.(5). If the density operators on the right-hand side of Eq.(5) are normalized, then the total density operator on the left-hand side is not normalized. Our reply We in fact had limited ourselves to the one-dimensional case. For larger dimensions, even for a small-scale cluster, the required calculational resources are currently unavailable. Therefore, most of the work in this area is focused on the one-dimensional case. We have added a more detailed description of the calculation scheme to the revised manuscript (Version 2) after Eq.(6). As regards the comment on the normalization of the right and left-hand sides of Eq.(5), we are grateful to the Referee for a useful remark. We corrected the typo in the formula, of course, in the actual calculation, the normalization is taken into account.
[1] H. B. Fei, B. M. Jost, S. Popescu, B. E. A. Saleh, and M. C. Teich, Entanglement-Induced Two-Photon Transparency, Phys. Rev. Lett. 78, 1679 (1997), publisher: American Physical Society. [2] I. Sinaysky, F. Petruccione, and D. Burgarth, Dynamics of nonequilibrium thermal entanglement, Phys. Rev. A 78, 062301 (2008). [3] J. Dajka, M. Mierzejewski, and J. Luczka, NonMarkovian entanglement evolution of two uncoupled qubits, Phys. Rev. A 77, 042316 (2008). [4] M. Urbaniak, S. B. Tooski, A. Ramsak, and B. R. Bul ka, Thermal entanglement in a triple quantum dot system, Eur. Phys. J. B 86, 505 (2013). [5] Z. Wang, W. Wu, and J. Wang, Steady-state entanglement and coherence of two coupled qubits in equilibrium and nonequilibrium environments, Phys. Rev. A 99, 042320 (2019). [6] T. Seidelmann, F. Ungar, A. Barth, A. Vagov, V. Axt, M. Cygorek, and T. Kuhn, Phonon-Induced Enhancement of Photon Entanglement in Quantum Dot-Cavity Systems, Phys. Rev. Lett. 123, 137401 (2019). [7] A. Ghannadan and J. Strecka, Magnetic-FieldOrientation Dependent Thermal Entanglement of a Spin-1 Heisenberg Dimer: The Case Study of Dinuclear Nickel Complex with an Uniaxial Single-Ion Anisotropy, Molecules 26, 3420 (2021). [8] V. E. Korepin, Universality of Entropy Scaling in One Dimensional Gapless Models, Phys. Rev. Lett. 92, 096402 (2004). [9] S. J. Gu, S. S. Deng, Y. Q. Li, and H. Q. Lin, Entanglement and quantum phase transition in the extended Hubbard model, Phys. Rev. Lett. 93, 086402 (2004). [10] B.Q. Jin and V. E. Korepin, Localizable entanglement in antiferromagnetic spin chains, Phys. Rev. A 69, 062314 (2004). [11] Y. Xu, H. Katsura, T. Hirano, and V. E. Korepin, Entanglement and Density Matrix of a Block of Spins in AKLT Model, J. Stat. Phys. 133, 347 (2008). [12] A. R. Its and V. E. Korepin, The Fisher-Hartwig Formula and Entanglement Entropy, J. Stat. Phys. 137, 1014 (2009). [13] C. C. Chang, R. R. P. Singh, and R. T. Scalettar, Entanglement properties of the antiferromagnetic-singlet transition in the Hubbard model on bilayer square lattices, Phys. Rev. B 90, 155113 (2014), publisher: American Physical Society. [14] F. Iemini, T. O. Maciel, and R. O. Vianna, Entanglement of indistinguishable particles as a probe for quantum phase transitions in the extended Hubbard model, Phys. Rev. B 92, 075423 (2015). [15] O. Vafek, N. Regnault, and B. A. Bernevig, Entanglement of Exact Excited Eigenstates of the Hubbard Model in Arbitrary Dimension, SciPost Phys. 3, 043 (2017). [16] F. Parisen Toldin and F. F. Assaad, Entanglement hamiltonian of interacting fermionic models, Phys. Rev. Lett. 121, 200602 (2018). [17] V. K. Vimal and V. Subrahmanyam, Quantum correlations and entanglement in a Kitaev-type spin chain, Phys. Rev. A 98, 052303 (2018). [18] F. Sugino and V. Korepin, Renyi entropy of highly entangled spin chains, Int. J. Mod. Phys. B 32, 1850306 (2018). [19] C. Walsh, P. Semon, D. Poulin, G. Sordi, and A.-M. S. Tremblay, Local entanglement entropy and mutual information across the mott transition in the two-dimensional Hubbard model, Phys. Rev. Lett. 122, 067203 (2019). [20] J. Spalding, S.-W. Tsai, and D. K. Campbell, Critical entanglement for the half- lled extended Hubbard model, Phys. Rev. B 99, 195445 (2019). [21] I. Kleftogiannis, I. Amanatidis, and V. Popkov, Exact results for the entanglement in 1D Hubbard models with spatial constraints, J. Stat. Mech: Theory Exp. 2019, 063102 (2019). [22] P. Padmanabhan, F. Sugino, and V. Korepin, Quantum phase transitions and localization in semigroup Fredkin spin chain, Quantum Inf. Process. 18, 69 (2019). [23] G. Mathew, S. L. L. Silva, A. Jain, A. Mohan, D. T. Adroja, V. G. Sakai, C. V. Tomy, A. Banerjee, R. Goreti, A. V. N., R. Singh, and D. Jaiswal-Nagar, Experimental realization of multipartite entanglement via quantum Fisher information in a uniform antiferromagnetic quantum spin chain, Phys. Rev. Res. 2, 043329 (2020). [24] S. Fraenkel and M. Goldstein, Symmetry resolved entanglement: exact results in 1D and beyond, J. Stat. Mech: Theory Exp. 2020, 033106 (2020). [25] C. Kokail, B. Sundar, T. V. Zache, A. Elben, B. Vermersch, M. Dalmonte, R. van Bijnen, and P. Zoller, Quantum variational learning of the entanglement hamiltonian, Phys. Rev. Lett. 127, 170501 (2021). [26] D. L. B. Ferreira, T. O. Maciel, R. O. Vianna, and F. Iemini, Quantum correlations, entanglement spectrum, and coherence of the two-particle reduced density matrix in the extended Hubbard model, Phys. Rev. B 105, 115145 (2022). [27] E. Saitoh, S. Okamoto, K. T. Takahashi, K. Tobe, K. Yamamoto, T. Kimura, S. Ishihara, S. Maekawa, and Y. Tokura, Observation of orbital waves as elementary excitations in a solid, Nature 410, 180 (2001). [28] J. Schlappa, K. Wohlfeld, K. J. Zhou, M. Mourigal, M. W. Haverkort, V. N. Strocov, L. Hozoi, C. Monney, S. Nishimoto, S. Singh, A. Revcolevschi, J.-S. Caux, L. Patthey, H. M. Rønnow, J. van den Brink, and T. Schmitt, Spin orbital separation in the quasi-onedimensional Mott insulator Sr2CuO3, Nature 485, 82 (2012). [29] Z. Nussinov and J. van den Brink, Compass models: Theory and physical motivations, Rev. Mod. Phys. 87, 1 (2015). [30] D. I. Khomskii and S. V. Streltsov, Orbital E ects in Solids: Basics, Recent Progress, and Opportunities, Chem. Rev. 121, 2992 (2021). [31] A. Oles, Fingerprints of spin-orbital entanglement in transition metal oxides, J. Phys. Condens. Matter 24, 313201 (2012).

---

## Round 2 · List of Changes

• We have expanded the Abstract section by significantly increasing the review of the entangled states (in particular for Hubbard model), and other cutting-edge research on temperature entanglement.
• We have added more detailed explanation of the importance of the model in hand (in the Conclusions section).
• In the Methods section, we have noticeably extended the description of our computational procedure, detailing each step and highlighting some of the key features.
• We have extended the section Conclusions. Here we explain in more detail the importance of the obtained results on the thermodynamics of entangled states in spin-pseudospin systems.
• We have corrected some misprints.

---

## Editorial Decision

published